# A Neurophysiological Perspective on a Preventive Treatment against Schizophrenia Using Transcranial Electric Stimulation of the Corticothalamic Pathway

**DOI:** 10.3390/brainsci7040034

**Published:** 2017-03-28

**Authors:** Didier Pinault

**Affiliations:** 1INSERM U1114, Neuropsychologie Cognitive et Physiopathologie de la Schizophrénie, Strasbourg F-67085, France; pinault@unistra.fr; Tel.: +33-(0)3-6885-3245; 2Unistra, Université de Strasbourg, Strasbourg F-67000, France; 3Fédération de Médecine Translationnelle de Strasbourg (FMTS), Faculté de Médecine, Strasbourg F-67085, France

**Keywords:** animal model, gamma frequency oscillations, glutamate, ketamine, network synchrony, *N*-*methyl*-d-aspartate, psychosis, sleep spindles, thalamic reticular nucleus, thalamus

## Abstract

Schizophrenia patients are waiting for a treatment free of detrimental effects. Psychotic disorders are devastating mental illnesses associated with dysfunctional brain networks. Ongoing brain network gamma frequency (30–80 Hz) oscillations, naturally implicated in integrative function, are excessively amplified during hallucinations, in at-risk mental states for psychosis and first-episode psychosis. So, gamma oscillations represent a bioelectrical marker for cerebral network disorders with prognostic and therapeutic potential. They accompany sensorimotor and cognitive deficits already present in prodromal schizophrenia. Abnormally amplified gamma oscillations are reproduced in the corticothalamic systems of healthy humans and rodents after a single systemic administration, at a psychotomimetic dose, of the glutamate *N*-*methyl*-d-aspartate receptor antagonist ketamine. These translational ketamine models of prodromal schizophrenia are thus promising to work out a preventive noninvasive treatment against first-episode psychosis and chronic schizophrenia. In the present essay, transcranial electric stimulation (TES) is considered an appropriate preventive therapeutic modality because it can influence cognitive performance and neural oscillations. Here, I highlight clinical and experimental findings showing that, together, the corticothalamic pathway, the thalamus, and the glutamatergic synaptic transmission form an etiopathophysiological backbone for schizophrenia and represent a potential therapeutic target for preventive TES of dysfunctional brain networks in at-risk mental state patients against psychotic disorders.

## 1. Introduction

Neurobiological disorders of the brain are an immense burden with a rising cost in our societies [1,2]. In the European Union, about a third of the total population suffers from mental disorders [2], and we are still missing effective treatments, free of side-effects, against complex neuropsychiatric illnesses such as schizophrenia.

Schizophrenia is a progressive, debilitating mental illness characterized by a loss of contact with reality, personality disorders, mood symptoms, sensorimotor and cognitive impairments (DSM-5). This clinical disorganization and cognitive deficits are associated with abnormal anatomo-functional connectivity and disturbances in neural oscillations and synchrony in highly-distributed brain networks, including thalamus-related circuits [3,4,5,6,7,8,9,10,11]. Schizophrenia has a multifactorial etiology involving genetic, neurodevelopmental, environmental and socio–cultural factors [12,13,14,15]. Therefore, the neurobiology of schizophrenia remains elusive. Its complex and multifactorial symptomatology has been driving multiple lines of research with diverse (genetic, immunological, metabolic, neurochemical, neurophysiological) hypotheses.

Most of the patients suffering from schizophrenia are treated with a combination of non-biological therapies (cognitive remediation, psychotherapies) and antipsychotic medications. These drugs are more effective against the positive than the negative symptoms of schizophrenia, and they induce serious adverse effects [16,17]. Because schizophrenia is characterized by multiple etiopathophysiological factors, does it make sense to have a treatment based principally on a “single receptor”? There is increasing evidence that the psychiatric disorders and the cognitive deficits of patients with schizophrenia are associated with and may be caused by dysfunction of highly-distributed systems displaying disturbed spatiotemporal activity patterns. Moreover, a recent meta-analysis demonstrated significant changes in whole brain network architecture associated with schizophrenia [18]. This supports the notion of functional interactions between multiple molecular, cellular and system pathways, which contribute to the multiple symptoms and cognitive deficits in schizophrenia [19]. So, would it not be more promising to use a therapeutic means that non-specifically modulates the ongoing global brain activity such as to re-establish normal brain function? The rationale behind the use of a non-specific therapeutic means in patients with schizophrenia is that the return to their normal brain function may especially alleviate the psychiatric disorders and dampen their emotion, sensorimotor and cognitive deficits. Deep brain stimulation (DBS) may be a promising alternative [20]. The challenge is immense given the complex symptomatology and pathophysiological mechanisms of schizophrenia (see below).

Davidson and colleagues [21] wrote: “To achieve the best therapeutic results in schizophrenia—like most other disorders—primary prevention is preferable to early and prompt treatment, which, in turn, is preferable to treatment of chronically established illness”. This raises the questions when to apply the preventive treatment and what could be the appropriate treatment modality? Ideally, assuming that schizophrenia is caused by an aggressive agent (microbe, virus, parasite) during pregnancy, it would be great to have the infection type identified to proceed to the appropriate preventive asymptomatic treatment [22,23]. Moreover, there is increasing evidence suggesting that maternal immune stimulation during pregnancy can increase the risk of neurodevelopmental disorders such as schizophrenia [24,25,26]. Prenatal infection may lead to a developmental neuroinflammation, which would contribute to the disruption of the normal brain development leading to dysfunctional networks and abnormal behavior relevant to schizophrenia [27]. The corresponding neuroimmune and behavioral abnormalities might occur in response to stress in puberty [27,28]. Rodent models of maternal immune activation mimic both behavioral and neurobiological abnormalities, which are relevant to schizophrenia [28,29,30,31]. Interestingly, an early presymptomatic anti-inflammatory intervention during peripubertal stress exposure can prevent the schizophrenia-relevant behavioral and neurobiological abnormalities [28]. Nevertheless, there is a real need to treat high-risk patients for whom the causes of their mental state remain unknown. That is why in the present review, I will present a neurophysiological perspective on a preventive symptomatic treatment in patients with premorbid and/or prodromal manifestations and bioelectrical markers of latent schizophrenia.

A universal property of brain networks is to produce electric currents, conveyed by the movement of ions across cellular membranes, whatever the pathophysiological context may be. Moreover, both the normal and the unhealthy brain can generate normal and abnormal brain rhythms [32]. Importantly, almost every cognitive task is associated with event-related electroencephalographic (EEG) oscillations [33]. We will see that disturbances in brain rhythms are common in patients with schizophrenia. So, can exogenous electricity correct or re-normalize abnormal brain oscillations and, in parallel or as a result, the related mental, emotional, sensorimotor and cognitive disorders? Numerous clinical interventions have demonstrated the benefits of brain electrical stimulation methods in patients with neurobiological disorders resistant to available pharmacological treatments.

There is increasing evidence that exogenous electric currents can modulate not only brain electrical activity but also behavioral and cognitive performance, giving hope for treating complex neuropsychiatric illnesses. Moreover, schizophrenia patients with auditory hallucinations that are unresponsive to antipsychotic medications can be treated with transcranial magnetic stimulation (TMS) [34,35] or transcranial electric stimulation (TES) [36,37,38,39,40]. Ongoing research aims to develop such noninvasive neurophysiological therapies as a routine therapy [40,41,42], which may be more promising than DBS (see below). The exponents of noninvasive technologies now face a huge challenge to develop and refine an efficient therapeutic, free of side-effects, neurophysiological method that treats schizophrenia in its entirety, that is, to treat all its core features, including positive, negative and mood symptoms, and the decline in cognitive abilities (memory and thinking skills). Although there is accumulating evidence that TES techniques are reliable and versatile therapeutic options, a certain number of questions remains open regarding their anatomical targets and the neural mechanisms underlying their clinical impact.

The development of the chronic character of schizophrenia takes years following the occurrence of prodromal symptoms with cognitive declines and first-episode psychosis [43,44,45]. Longitudinal studies in at-risk mental state individuals and the duration of untreated psychosis offer a time window to identify predictive biomarkers, to better understand the etiopathophysiology of schizophrenia and to develop innovative therapies [44]. It would be ideal to have a therapeutic neurophysiological modality, for instance, a weak TES (see below) applied in at-risk individuals, capable of stopping the occurrence of first-episode psychosis and chronic schizophrenia. This exciting idea has already received increasing interest during at least the last two decades [43,46,47,48].

Electro- and magneto-encephalographic oscillations are natural bioelectrical markers of the functional and dysfunctional state of brain networks [4,49]. Spontaneously-occurring gamma frequency (30–80 Hz) oscillations (GFO) of cortical origin, naturally implicated in attention and integration processes, are excessively amplified not only during hallucinations [50,51,52,53] but also in first-episode psychosis and in at-risk mental states for psychosis [54,55,56,57]. Such abnormally amplified GFO can be reproduced after a single systemic administration, at a psychotomimetic dose, of the glutamate *N*-*methyl*-d-aspartate receptor (NMDAR) antagonist ketamine in the corticothalamic (CT) systems of healthy humans [58] and rodents [59,60,61,62]. These translational models of first-episode psychosis highlight one important factor, glutamate synaptic transmission, which may be altered in individuals at high risk of developing psychotic disorders [63]. Also, disturbances in sleep and reductions in EEG sleep spindles in first-episode psychosis and early-onset schizophrenia [3,64,65,66,67] support the hypothesis that the thalamus plays a critical role in the pathogenesis of schizophrenia.

Therefore, in the present essay, I put forward the notion that the CT pathway, the thalamus, and glutamate synaptic transmission might together represent the backbone etiopathophysiological mechanism of chronic psychotic disorders. This mainstay mechanism may be a prime target for early therapeutic intervention using TES techniques, more specifically targeting the CT pathway. First of all, I will start with a discussion about possible anatomical targets for late therapeutic electrical stimulation in patients with advanced schizophrenia. Then, I will provide an overview of therapeutic neurophysiological procedures and will develop a theoretical proposal on how we may understand the mechanisms underlying the effects of TES of the cerebral cortex with a focus on the CT pathway. It is worth specifying that, in the present survey, the rodent is our predilection animal since its neural networks share common anatomo-functional properties with those of humans.

## 2. Is There an Anatomical Target for Advanced Schizophrenia?

It is extremely challenging to find the correct or best anatomical target(s) for the use of invasive or non–invasive electrical stimulation of brain networks as last resort treatment of complex mental-health disorders, such as schizophrenia. For instance, low-frequency TMS of the left temporoparietal cortex can reduce positive symptoms, especially self-reported auditory hallucinations [34,68,69]. Although there is evidence that TMS can enhance cortical synchrony, improve cognitive performance [70,71], and is safe and free of cognitive side-effects, further investigations are however required to identify both the anatomical target(s) and the stimulation settings that would lead to an efficient treatment against both positive and negative symptoms of schizophrenia [72,73].

In contrast, applying DBS at the seemingly best anatomical target might not be free of side-effects. For instance, it is well known that DBS treatment in the subthalamic nucleus can not only alleviate essential and Parkinson disease tremors but also reduce symptoms in patients suffering from severe forms of obsessive compulsive disorders, providing encouraging findings that are corroborated by animal studies [74]. Even if the subthalamic nucleus might be an effective target for the treatment of behavioral disorders that include emotional, cognitive, and motor impairments [75], its use in patients experiencing severe psychiatric disorders with limited therapeutic options remains questionable.

Patients who severely suffer from advanced psychiatric disorders, and who are refractory to medication, must benefit from a new efficient therapeutic approach, and DBS may be a promising alternative. Goodman and Insel [20] recently put emphasis on the fact that the pace of DBS being used against seriously advanced neuropsychiatric disorders is accelerating, giving the incentive both basic and clinical researchers need to concentrate their efforts on the road ahead. However, are we actually ready to use DBS against severe and ultra-complex mental-health disorders? Schizophrenia, as a typical example, is a multidimensional and multifactorial illness, raising an important question as to whether or not DBS could alleviate both the negative and the positive symptoms in all schizophrenic patients who urgently need a new effective therapy [76]. Could both the anatomical target and the DBS settings used in any specific patient be also generally applied to other very problematic patients? The challenge is immense because of the existence of several types of schizophrenia [77]. Finding a unique effective therapy against all types of schizophrenia presupposes that they all share common etiopathophysiological mechanisms. In psychiatry, DBS may be effective in combination with commendable clinical practices [78,79]. Therefore, the use of non–human animal and network models remain a versatile means to develop therapeutic concepts and to understand the neural mechanisms of electrical neuromodulation used in diverse interventions.

Reliable and reproducible non-human animal models of schizophrenia do not exist, and any model for schizophrenia remains questionable with its strengths and limitations [80,81,82,83]. Indeed, schizophrenia belongs to a group of hyper-complex mental-health disorders. So, how to model the heterogeneity of the causes, the progression, the multiple clinical symptoms of chronic schizophrenia, and of the changes elicited by years of medication? The critical problem in finding an efficient treatment of schizophrenia is due to the challenge in modeling psychiatric disorders, which depends on the lack of information about their etiology and pathophysiology.

So, what would be a “good” model of or for advanced schizophrenia (untreatable using the currently available therapeutic means) suitable to develop a therapeutic concept based on the use of invasive or noninvasive electrical stimulation of the appropriate neural networks? In theory, one may believe that an animal model that is not validated as being a good model for schizophrenia but that is validated as being a good model for a measurable, singular pathophysiological behavioral trait (e.g., violent behavior, catatonia) similar to that observed in advanced schizophrenic patients and that often makes it to the headlines of newspapers [84,85,86], may have an added value to investigate the efficacy of a therapeutic treatment using DBS in clinical trials. Again, validation of such a concept (reversal of the behavioral/pathophysiological trait by DBS) should rest on well identified anatomo-pathophysiological mechanisms, and they should be conducted along with appropriate ethical guidelines.

Animal models are, however, all potentially useful as long as they are precisely described, and as the related working hypotheses are clearly formulated. This critical view is still a matter of discussion [80,81,83,87,88]. So, along these lines, any model can be a versatile tool to explore the multiple facets concealed by healthy and sick brains. The challenge is indeed to find convergences across models and patients, at least in terms of symptoms and neural dysfunction. Then comes the question of how animal models can be used to discover the appropriate therapy? Neurodevelopmental rodent models, based on prenatal immune activation, which present at the adult stage schizophrenia-relevant behavioral and neurobiological abnormalities [89,90,91], may be promising. Indeed, it was recently demonstrated that a preventive, presymptomatic anti-inflammatory treatment during peripubertal stress exposure can prevent the abnormal behavior and the biomarkers of the neuroimmune abnormality [28]. However, there is a real need to find an appropriate treatment of psychosis for high-risk patients for whom the causes of their mental state remain unknown.

Also, if we have known the etiological cause(s) of a given patient suffering from schizophrenia for many years, would we be able to find the appropriate treatment that alleviates simultaneously the positive and negative symptoms and the cognitive and emotional disorders? For the time being, we face the absolute necessity to understand the etiology and pathophysiology of schizophrenia at the molecular, cellular and systems levels, with the dream to find an effective, asymptomatic or symptomatic treatment free of side-effects.

## 3. Overview of Therapeutic Neurophysiological Procedures: DBS versus TES

Since the middle of the 20th century, invasive and noninvasive neurophysiological approaches have been attracting increasing interest as means of last resort treatments against advanced neurological and mental illnesses that are resistant to currently available therapies. Deep brain electrical stimulation has evolved as an invasive, stereotaxic-guided [92] and neuroimaging-guided [93] neurophysiological treatment when drug therapy no longer provides relief from symptoms accompanying severe neurological and psychiatric disorders. It is now used to treat other severe brain disorders in patients resistant to pharmacological mono- and polytherapy, including Parkinson’s disease [94,95] epilepsy [96], dystonia [97], obsessive-compulsive disorders [74], pain [98,99], multiple sclerosis [100], depression [101,102] and Tourette syndrome [103]. It is also used in brain-injured patients in vegetative and minimally conscious states [104]. Although therapeutic DBS is applied along with the rules of the art and ethics [105], its use can be accompanied by psychiatric complications [106,107]. The neural mechanisms underlying the effects of DBS are complex and little known [108,109].

Since the end of the 20th century, TMS has emerged as a tool to study the human brain and an efficient noninvasive therapeutic means against depression [110], schizophrenia [34,35], addiction [111] and other neurological and psychiatric disorders [42]. However, there is a need to optimize the TMS protocols for routine clinical practice. TMS excites or inhibits the activity of cortical nerve cells and the dynamics and plastic properties of neural networks through the influence of electric currents that are induced by changing magnetic fields. Repetitive TMS modulates in a frequency-dependent manner the excitability of the cortical circuits [112]. Regarding schizophrenia, although TMS is efficient in treating auditory hallucinations [34,68,69], a major issue is to find the anatomical target and the TMS settings that allow treating the disease in its entirety, that is, to alleviate all the negative and positive symptoms, mood disorders and cognitive deficits.

Since the beginning of the 21st century, three principal TES techniques, forms of noninvasive and less aggressive neurophysiological modulation, have been increasingly used in cognitive neurosciences and interventional approaches [37,40]: transcranial direct current stimulation (tDCS), alternating current stimulation (tACS), and random noise stimulation (tRNS). Transcranial electrical stimulations safely apply, via scalp electrodes, a weak electrical current to modulate the physiological or pathological cortical and subcortical activities of healthy subjects or patients suffering from severe mental disorders [113]. Such TES techniques can modulate synaptic plasticity and related genes [114]. It seems not yet clear whether the application of TES on the frontal cortex of patients with schizophrenia can bring significant beneficial effects [115]. However, tDCS applied over the left frontotemporal cortex of schizophrenia patients with disabling treatment-resistant symptoms reduces both the auditory verbal hallucinations (~30%) and the abnormal resting-state hyperactivity between the left temporoparietal junction and the anterior insula [116]. When applied to the temporoparietal cortex of schizophrenia patients with medication-refractory auditory verbal hallucination, tDCS can not only decrease the severity of the hallucinations but also ameliorate some negative and positive symptoms [117]. tDCS is a static current that polarizes the membrane potential of the neuronal elements of the target cortical volume [118,119] whereas tACS modulates, in a frequency specific manner (within the EEG frequency range), ongoing cortical neural oscillations [120,121]. There is increasing evidence that tDCS can induce memory enhancement in healthy subjects, in patients with psychiatric and neurological disorders, and in animal models [122]. In contrast to tDCS, tACS can modulate, more directly, not only the firing of the nerve cells but also their oscillations and synchrony [40]. In a subpopulation of patients with schizophrenia, tDCS can be efficient in the reduction of refractory verbal hallucinations but also of positive and negative symptoms [116,117]. Interestingly, an imperceptible alternating current (peak-to-peak: 750 μA) applied at the gamma frequency (40 Hz) to the frontal cortex can enhance cognitive performance during logical reasoning [123]. Also, gamma frequency (25–40 Hz) tACS applied on the frontotemporal cortex of subjects during REM sleep influences ongoing brain activity and induces conscious awareness, making it possible for the dreamer to be lucid of his or her dream and to have control of its content [124]. Gamma frequency oscillations are well known to be prevalent during REM sleep [125]. It was recently shown that gamma tACS of the human motor cortex increases motor performance during a visuomotor task [126]. Concurrent functional magnetic resonance imaging has revealed neural activity changes underneath the stimulation electrode and in related brain networks, including the prefrontal cortex [126]. On the other hand, alpha frequency (10 Hz) tACS applied over the occipitoparietal cortex reduced cognitive performance in a visual task [127]. The effects of tACS on brain network oscillations and on behavior are critically discussed in a comprehensive review article [128]. tRNS is an alternating current stimulation technique with a wide-band stimulation frequency (up to 640 Hz) [129]. It has been shown to increase neuroplasticity during perceptual learning [130].

In summary, TES techniques appear promising for clinical interventions. They are safer than DBS techniques and less expensive than TMS. TES is also more appropriate than DBS as a preventive treatment modality against schizophrenia because it is noninvasive and almost free of side-effects. In addition, the efficacy of TES techniques to modulate brain activities and to influence cognitive performance have been demonstrated. The mechanisms underlying their effects, however, remain still elusive. Also, more research and clinical trials are necessary to attain, during routine clinical practice, consistent benefits for patients suffering from debilitating mental illnesses. Importantly, emerging clinical interventions have shown that TES therapeutic modalities can reduce essential tremors in patients suffering from Parkinson’s disease [131,132]. So, one can easily imagine that the noninvasive and low-cost TES techniques might supplant invasive DBS methods.

## 4. Three Candidates for Preventive TES

As mentioned above, it is extremely challenging to make a decision for a late therapeutic neurophysiological intervention in treatment-resistant advanced schizophrenia. The development of chronic forms of schizophrenia takes years after the occurrence of prodromal symptoms, cognitive declines and first-episode psychosis [45,133,134,135,136,137,138]. So, would it be possible to prevent or to delay the progressive development of chronic schizophrenia? Early therapeutic intervention is a notion that has been around with increasing interest during the last two decades [44,46,47,139]. Indeed, the complex symptomatology of schizophrenia results from progressive abnormalities of brain networks, including the thalamus with its reciprocal connections with the cerebral cortex [11,140,141,142]. Even if cognitive impairments are relatively modest during the prodromal phase of schizophrenia [137,143], efficient early therapeutic intervention could stop or delay the onset of psychotic disorders, which might otherwise lead to further cognitive damage and impaired daily functioning [144,145,146].

A preventive treatment against schizophrenia of course requires a better understanding of the evolution of the clinical disorganization and of the cognitive changes that can be observed from premorbid to first-episode psychosis [136,143,147]. The great challenge is to identify the time window when the very first clinical symptoms and cognitive declines start to occur in individuals who will actually develop schizophrenia [143]. The “primary” factor(s) responsible for the progressive neural changes leading to chronic forms of schizophrenia remain to be identified. Whatever the preventive neurophysiological therapy implies, it should target an etiopathophysiological mechanism that is at the root of the mental disorders. Multiple diverse (genetic, epigenetic, neurodevelopmental, immunological, environmental, socio-cultural) factors are thought to be (either in isolation or through interactions), the cause of schizophrenia [14,148,149,150,151,152,153,154,155]. This is a long–standing debate that is not discussed in the present essay. Here, I highlight recent findings supporting the glutamate neurotransmission hypothesis of schizophrenia, which implicates critical etiopathophysiological mechanisms that appear early during its progressive development. Here, it is assumed that these processes are common to several types of chronic schizophrenia during the prodromal phase.

Glutamate is one of the main neurotransmitters of the thalamus, an essential subcortical structure involved in sensory discrimination, motor and cognitive processes. It is worth mentioning that more than 80% of the thalamic neurons are glutamatergic and they are massively innervated by the CT neurons, which are also glutamatergic [156]. Importantly, there is compelling evidence that multiple and diverse abnormalities (glutamate receptors, transporters and associated proteins; NMDAR–associated intracellular signaling proteins, and glutamatergic enzymes) related to the glutamate transmission have been found in the thalamus of patients with schizophrenia [157]. Furthermore, in the following, we describe compelling evidence that the pathogenesis of first–episode psychosis can be better modeled translationally than chronic forms of schizophrenia. Therefore, we will also demonstrate that the thalamus might be an interesting target for TES, directly and indirectly via the glutamatergic CT pathway, designed for early therapeutic intervention against first-episode psychosis and chronic forms of schizophrenia.

### 4.1. The Corticothalamic Pathway and the Thalamus

The thalamus, located around the third ventricle, is reciprocally connected with the cerebral cortex and it receives inputs from the cerebellum, the basal ganglia, the brainstem and basal forebrain [158]. The thalamus is an essential relay and plays an integrative role for ongoing and function-related cortical activities [159]. It relays information to multiple regions of the cerebral cortex in a bottom-up (from the external world via the sensory receptors) and a top-down (from the cerebral cortex) fashion. During sensory discrimination, sensorimotor integration and cognitive processes, the information circulates along the glutamatergic CT and thalamocortical (TC) pathways [160]. The thalamus is implicated in multiple functions: sensory perception (visual, somatosensory, auditory), sleep, wakefulness (through the ascending activating system), pain, attention and consciousness [156]. It is also implicated in many neurological and psychiatric diseases, including Alzheimer’s disease, Parkinson disease, epilepsy, schizophrenia, autism, bipolar disorders, chronic pain and major depression. Damage to the thalamus can cause very long-lasting (>3 years) or permanent coma [161,162]. The thalamus is also the anatomical target of therapeutic DBS methods [104,163,164]. The specific prethalamic (or peripheral) inputs of the sensory systems innervate both the specific or first-order (primary sensory) and the non-specific or higher-order (associative and cognitive) thalamic nuclei. First- and higher-order thalamic nuclei relay information to the granular and the supra- and infragranular layers of the neocortex, respectively [160].

TC neurons, the principal neurons of the thalamus, are glutamatergic and their axon relays thalamically processed signals to the cerebral cortex (Figure 1A). In contrast to the CT pyramidal neurons, the TC neurons do not communicate with each other. The TC axons give rise to en passant collaterals in the GABAergic thalamic reticular nucleus (TRN)—a thin layer that covers the lateral walls of the dorsal thalamus—which is the principal source of GABA receptor-mediated inhibition of TC neurons [165]. The TC-related information is also computed in vertical (within the cortical column) and horizontal (between columns) cortical networks, linked with other cortical (distant areas) and subcortical structures (e.g., striatum, amygdala, and hippocampus; Figure 1B). Intracortically-computed information reaches the thalamus via CT axons. Thereby, both the thalamus and the cerebral cortex work together in concert through their topographically organized reciprocal connections, which form intermingled feed-forward and closed-loop CT-TC circuits [156,166]. The CT pathways are of two types, one originating in layer V and the other in layer VI.

The GABAergic TRN is an inhibitory interface strategically located between the thalamus and the neocortex [165,167]. It is innervated by glutamatergic TC and layer VI CT inputs (Figure 1 and Figure 2). TRN cells have dendro-dendritic synapses to communicate among each other [168,169,170,171] (Figure 2). The TRN is also characterized by an important intrinsic network of chemical (GABAergic) and electrical synapses [172,173], which can effectively be influenced by the layer VI CT pathway [174,175,176]. The hodology and the innervation pattern of the CT-TRN-TC circuit strongly indicate that the GABAergic TRN neurons are implicated in both top-down and bottom-up processing, suggesting that the TRN might play a central role in attentional processes [165]. Moreover, lesions of the TRN lead to attentional deficit [177,178]. Their axonal projections are topographically organized and form open-loop connections with the related TC neurons, the anatomical substrate of lateral inhibition in the thalamus [179,180]. Thereby, TRN neurons can modulate, in a coordinated fashion, the TC activities that are relevant for attention and integration processes. In schizophrenia, disorders of thalamic lateral inhibition are thought to disturb the pattern of TC activity on the way to the cerebral cortex [8,181]. Importantly, GABAergic TRN neurons are endowed with powerful oscillatory properties (see below).

The layer V CT pathway selectively innervates the higher-order thalamic nuclei in a focal manner (like a driver input). Like the peripheral inputs, it does not innervate the TRN, in contrast to the layer VI CT pathway. This layer V CT pathway is an essential element in cortico-cortical (or transthalamic) circuits, which parallel direct cortico–cortical connections [159]. The principal axon of layer V CT neurons also innervates motor centers in the brainstem and spinal cord. The axonal branch that innervates the thalamus conveys corollary discharges used to modulate imminent sensorimotor processing [159,182,183]. In fact, corollary discharges might be disturbed in schizophrenia [184,185,186]. Both layer V and VI CT pathways are assumed to work together in synergy from the very first stages of sensorimotor processing up to subsequent higher cognitive and motor processes.

The layer VI CT pathway plays an essential role in attentional and integrative processes [8,187,188,189]. This CT pathway innervates the TRN and the related first- and higher-order thalamic nuclei [156,159]. This pathway exerts a massive (about ten–fold stronger than the corresponding TC pathway) [190] and regional innervation within large thalamic territories (Figure 2). Cortical layer VI contains a heterogeneous population of neurons [191,192,193]. The layer VI CT pathway is the major glutamatergic output, which is reciprocally connected with TC neurons [188,194]. Layer VI CT apical dendrites and axon collaterals terminate in layer III–IV [188,193]. Their axon collaterals are implicated in both excitatory and inhibitory feedback mechanisms in layer IV [195,196]. Layer VI CT axons innervate other layer VI CT neurons [197,198]. Their apical dendrites can perform active integration of synaptic inputs via dendritic spiking [199]. There is anatomical evidence that some dendritic spines of neocortical pyramidal neurons are simultaneously innervated by GABAergic and glutamatergic inputs from local–circuit cells and TC neurons, respectively [200]. Thereby these GABAergic inputs can gate the synaptic impact of the incoming TC inputs on the pyramidal neurons. Layer VI CT neurons mediate most of their excitatory neuronal transmissions through the activation of ionotropic (NMDA and AMPA) and metabotropic glutamate receptors in both the cortex and the thalamus [201]. Interestingly, Layer VI CT neurons innervate not only the thalamus but also cortical layer IV, suggesting that layer VI CT neurons exert a dual, intrathalamic and intracortical, feedback control of incoming TC activities [159,202]. Such a cortical feedback would have a facilitatory effect on the thalamus [203]. Thereby, the spatiotemporal dynamics of intracortical synaptic and intrinsic processes, especially in layer VI dendrites, are under the influence of the dialogue between the corresponding CT and TC neurons.

In the thalamus, NMDAR-mediated excitatory postsynaptic currents are much larger in CT than in prethalamic (sensory inputs) synapses [204,205,206]. Importantly, the corresponding NMDAR-related response is antagonized by the NMDAR antagonist ketamine or MK-801 [207], which significantly increases the power, and the synchrony, of ongoing GFO in the highly-distributed CT-TC systems [59,60,61,62]. Moreover, the CT pathway significantly contributes to thalamic GFO [59,208]. The layer VI CT pathway also exerts a great influence on the state of the membrane potential of the TC neurons, as well as on ongoing and function-related thalamic activities. More specifically, the CT neurons shape the spatiotemporal receptive fields of TC cells [209,210] and play an essential role in the coordination of widespread coherent oscillations [211]. Importantly, the CT innervation, mediated by both NMDA and non-NMDA receptors [212], is more effective to the TRN than to TC neurons [174]. In the TRN, the CT pathway involves not only monosynaptic excitations but also disynaptic and polysynaptic GABA(A)-mediated inhibitions [176]. Thereby, the layer VI CT pathway and the TRN work together as an attentional searchlight (focused attention) to salient sensory stimuli [165,213,214,215]. There is a large body of comprehensive anatomo-functional studies showing that CT neurons exert a simultaneous effect on both the center (excitation) and the surround (suppressive) of receptive fields [189,215,216,217,218]. The CT synapses would thus exert a crucial role in sharpening the thalamic receptive field via intensifying both the center and the surround mechanisms. The CT influence is dynamic with an excitation-inhibition balance changing in an activity-dependent manner [205]. Sustained cortical activity enhances thalamic activities, such as during states of focused attention [219,220]. Finally, CT neurons are thought to function similarly across species and across sensory modalities [218].

Thalamic rhythms: The thalamus plays a crucial role in the generation of brain rhythms [221,222] and it is implicated in a wide range of brain oscillations [223,224,225]. Indeed, the thalamic neurons are endowed with state-, time- and voltage-dependent properties, under the control of synaptic inputs, which allow them to fire a single action potential or a high-frequency (up to 600 Hz) burst of two to seven action potentials. The firing mode, tonic or bursting, is determined by low–threshold T-type calcium channels. They are inactivated at a membrane potential more positive than −60 mV and de-inactivated below for more negative values. This means that for a membrane potential hyperpolarized below −60 mV, any intrinsic depolarization or depolarizing input, including a reversed inhibitory postsynaptic potential, can trigger a low–threshold potential crowned by a high-frequency burst of action potentials. In short, when the T channels are inactive, the thalamic neurons fire in a tonic manner; they fire in the burst mode when the T channels are de-inactivated. These electrophysiological properties have been characterized in detail in a countless number of publications (for review see, e.g., [226,227,228]).

The GABAergic TRN cells are also endowed with state- and voltage-dependent pacemaker properties not only at the spindle frequency (7–12 Hz) [229,230,231] but also at the gamma frequency [232] oscillations. Indeed, the membrane of the GABAergic TRN neurons can generate intrinsic subthreshold and threshold GFO, which result in rhythmic GABA(A) receptor-mediated inhibitory postsynaptic potentials in related TC neurons [232]. The oscillating properties of TRN and TC neurons are influenced by the CT pathway [59,205,208,233,234]. Therefore, the TRN plays a key role in the state-dependent generation of thalamic GFO, which are under the powerful control of large populations of layer VI CT neurons.

In schizophrenia, the thalamus and its related networks present diverse (structural, chemical, physiological and metabolic) abnormalities [5,7,11,140,235,236,237,238]. Volumetric and structural studies using imaging have revealed a reduction in the volume of the thalamus not only in chronic schizophrenia [239] but also in first-episode psychosis and in antipsychotic-naive high-risk individuals for psychosis [240]. These structural changes may be linked to a functional dysconnectivity between the thalamus and the cerebral cortex in both early and chronic stages of psychosis, which is associated with cognitive impairment [10,135,241,242,243]. A decrease of the thalamic glutamate level has also been measured [244,245], and almost all molecules implicated in the glutamate transmission pathway are altered in the thalamus of patients with schizophrenia (changes in the expression of glutamate receptors, transporters and associated proteins) [157]. The thalamic glutamate level, measured with the use of proton magnetic resonance spectroscopy, might also be a predictor of psychosis [244,245].

In first-episode and early–onset schizophrenia patients, disturbances in sleep represent a core pathophysiological feature. Cortical EEG studies conducted in such patients have revealed a significant deficit in sleep spindles, a marker of functional dysconnectivity [3,246]. This might be due to a reduced function of NMDAR, as an in vitro study, conducted in thalamic slices, demonstrated that a selective blockade of NMDAR or a diminished extracellular magnesium concentration significantly shortens spindle-like oscillations [247]. However, we do not know whether the reduction in sleep spindles in patients with schizophrenia is due to a presynaptic and/or postsynaptic dysfunction of TC or TRN neurons. It is tempting to speculate that the functional dysconnectivity recorded in schizophrenia also involves a reduction of NMDAR activity. This hypothesis is supported by the fact that, in rodents, the NMDAR antagonist ketamine, at a psychotomimetic dose, disrupts the functional state of the CT pathway [59].

In summary, the thalamic volume and glutamate level, sleep spindles and ongoing GFO are potentially useful biomarkers for the clinician to diagnose the prodromal phase of psychosis. Therefore, the thalamus with its structural, neurochemical and electrophysiological properties seems an essential structure in the etiopathophysiology of schizophrenia, as well as a prime target structure for preventive TES, directly and indirectly via the CT pathway.

### 4.2. Glutamatergic Transmission

In the light of our current knowledge, the term “glutamate hypothesis” mean that schizophrenia is caused by multiple variables and a stream of pathophysiological processes related to NMDAR–related synaptic functions [248,249]. NMDAR are well known to play, by means of synaptic plasticity, an essential role in the adequate neurodevelopment of cognitive abilities [250]. Here, the glutamate hypothesis does not negate the dopamine hypothesis and the other pathophysiological hypotheses of schizophrenia. Moreover, the disturbed dopaminergic and glutamatergic neurotransmissions might be causally related [44,251,252,253].

Glutamate is the predominant neurotransmitter in the brain. It is the precursor of GABA, the most prevalent inhibitory neurotransmitter that balances glutamate’s actions. Glutamate works with ion channel–associated (ionotropic) or G protein–coupled (metabotropic) receptors. It is also well known that NMDARs play, by means of synaptic plasticity, an essential role in the adequate neurodevelopment of cognitive abilities [250]. Since 1980, there have been increasing lines of evidence suggesting that glutamate-based synaptic neurotransmission is altered in schizophrenia [254,255,256]. Kim and colleagues (1980) measured a decrease of glutamate in the cerebrospinal fluid of an untreated patient with schizophrenia. Then, studies performed on postmortem human brain samples demonstrated changes in glutamate receptor binding, transcription and subunit protein expression in the prefrontal cortex and subcortical structures, including the thalamus and hippocampus [257]. They also showed altered levels of the amino acids *N*-acetyl aspartate (NAA) and *N*-acetyl aspartyl glutamate (NAAG) and of the activity of the enzyme that cleaves NAA to NAAG and glutamate in the cerebral spinal fluid and postmortem tissues from patients suffering from schizophrenia [258,259]. Also, genetic studies have revealed that a majority of genes associated with schizophrenia are linked to the glutamatergic system [248,260,261,262,263]. Interestingly, an imaging study (SPECT tracer for the NMDAR) revealed a reduction in NMDAR binding in the hippocampus of medication–free patients [264]. Even when considering the possibility that schizophrenia is caused by an immune dysfunction due to infectious agents, a link is identified between immune alterations and disturbances of glutamate NMDA receptors [153,265]. Interestingly, there is a growing body of findings indicating that glutamate synaptic transmission is significantly altered in schizophrenia since the premorbid phase [244,245,254,265,266,267].

A systemic single dose administration of non–competitive NMDAR antagonists (phencyclidine, ketamine or MK-801) elicits cognitive deficits and schizophreniform symptoms in healthy individuals and greatly exacerbates the symptoms in patients with schizophrenia [268,269,270,271,272,273,274]. The ketamine–induced schizophreniform symptoms are associated with a state of functional cortical–subcortical hyperconnectivity [275,276] and an abnormal amplification of baseline GFO, reminiscent of the increased GFO observed during hallucinations [50,51,52,53] and in at-risk mental state individuals for psychosis (untreated with ketamine) [56]. These clinical neurophysiological findings were predicted by comprehensive preclinical studies conducted in acute ketamine rodent models [60,62,277,278,279,280].

In summary, disturbances in glutamate synaptic transmission, involving a reduced function of NMDAR with multiple functional consequences, start to appear early during the development of schizophrenia. This may cause the dysfunctional neural plasticity in schizophrenia [281]. A certain number of genes (DISC-1, dysbindin, SHANK, and NRG-1) are well-known to modulate NMDAR-mediated glutamate transmission [282,283]. This notion is supported by patients with an autoimmune encephalitis because they produce antibodies against NMDAR and have a clinical disorganization that is similar to that of patients with schizophrenia [284]. Therefore, glutamate transmission appears a potential “primary” target for an early therapeutic intervention [285,286]. Of importance, the psychosis-relevant abnormal amplification of GFO is reliably reproduced in healthy humans and rodents under the acute influence of the NMDAR antagonist ketamine at a psychotomimetic dose [58,60]. So, these translational acute pharmacological models seem appropriate to develop an innovative preventive treatment against the development of chronic psychotic disorders. It is well recognized that reduced function of NMDAR is a crucial factor for the progression and symptoms of schizophrenia [284]. It is tempting to posit that an appropriate preventive treatment would correct the dysfunctional brain network plasticity.

## 5. Gamma Frequency (30–80 Hz) Oscillations, a Potential Pathophysiological and Therapeutic Bioelectrical Marker

In the present essay, I put emphasis on GFO because there is compelling evidence of functional links between GFO, NMDAR hypofunction and a reduction in the number and the function of cortical GABAergic interneurons in schizophrenia [287,288,289,290]. This implies that GFO are considered a common denominator of the above–presented three facets (CT pathway, thalamus, and glutamate transmission), which represent the etiopathophysiological backbone for premorbid, acute and chronic psychotic disorders. Indeed, (i) coherent GFO are recorded not only in the neocortex but also in the related thalamus; (ii) the layer VI CT pathway contributes to thalamic GFO; and (iii) GFO increase in amplitude and power not only during hallucinations, in at-risk individuals for psychosis, but also after the administration, at a psychotomimetic dose, of the NMDAR antagonist ketamine. It is worth remembering that, in humans, GFO start to emerge several months after birth [291]. It was demonstrated that, during rodent neural development, thalamic GFO play a crucial role in the mapping of the functional TC modules [292]. Both in humans and rodents, GFO are simultaneously present in cortex and thalamus [59,293]. In the following, we will see that GFO are also potential bioelectrical markers of psychosis, which could be used for the development of therapeutic interventions.

Coherent synchronized GFO emerge in large-scale cortical-subcortical networks spontaneously or during global brain operations such as attention, perception, and memory [294,295,296,297]. They are thought to play an essential role in synaptic plasticity [298], spatiotemporal coding (binding by synchronization), storage and recall of information [299,300,301,302,303]. Network GFO are ubiquitous and operate in combination with other brain rhythms [224,304,305]. Extracellular field GFO principally result from subthreshold, synaptic and intrinsic membrane potential oscillations that trigger action potentials at a precise phase during the oscillatory period. Their functions and mechanisms are still a matter of debate. The functional interactions between GABAergic and glutamatergic neurons are thought to be responsible for the generation of GFO during attention and integration processes [304,306].

There is accumulating evidence that, in schizophrenia, the dysfunctional network connectivity between cerebral cortex and thalamus is accompanied by disturbances in GFO and by deficits in sensorimotor and cognitive performance [4,53,307,308,309]. There is also evidence of a correlation between schizophrenia–related symptoms and in particular cognitive and perceptual deficits with disturbances in cortical GFO [310,311,312], also in first-episode schizophrenia [6]. Gamma oscillations may be considered not only as neurophysiological markers of the functional state of brain networks but also as trait markers in schizophrenia [313]. Of importance, abnormally increased GFO are recorded in patients with first-episode schizophrenia [6,54,55,314,315] and in at-risk mental state patients for psychosis. Gamma oscillations are also abnormally excessive in amplitude during hallucinations [50,51,52,53]. Increased GFO are associated more with positive (such as hallucinations) than negative symptoms [4,316]. Therefore, hypersynchronized GFO look like a predictive bioelectrical marker for both psychosis and treatment outcome.

As reported above, such abnormally amplified GFO can consistently be reproduced in healthy humans and rodents following the systemic administration at a psychotomimetic dose of the NMDAR antagonist ketamine [58,60,317]. These translational acute ketamine models, which model the prodromal phase of psychotic disorders and first–episode psychosis, are thus appealing to work out a preventive treatment against the occurrence of chronic forms of schizophrenia. It may be worth specifying that a single low-dose (<10 mg/kg) application of ketamine in rats increases hyperfrontality, which can also be observed in first–episode schizophrenia [275,276,318]. In contrast, hypofrontality is diagnosed in patients with chronic schizophrenia. Therefore, the acute ketamine model may be more appropriate to mimic the pathogenesis of acute psychotic states in humans [268,271,273,275,276,317].

Abnormally amplified GFO in neural networks may contribute to the disruption of the integration of task–relevant information, which is part of psychotic symptoms [55,319]. Moreover, in rodents, a single systemic administration (at a subanesthetic dose) of ketamine disturbs the functional state of the somatosensory CT-TRN-TC circuit (Figure 3). Ketamine reliably increases the amplitude and power of spontaneously-occurring GFO and decreases the amplitude of the sensory-evoked potential and its related evoked GFO in both the thalamus and the neocortex. In other words, the NMDAR antagonist ketamine generates persistent and generalized hypersynchrony in GFO, which act as an aberrant diffuse network noise under these conditions, and represent a potential electrophysiological correlate of a psychosis-relevant state [60]. Such a generalized network gamma hypersynchrony thought to create a hyper-attentional state (see discussion in [59]), might be the force that disrupts the flow of sensorimotor and cognitive processes as observed in schizophrenia. Thereby, ketamine reduces the ability of the somatosensory CT-TRN-TC system to encode and integrate incoming information, perhaps by disrupting the center-surround receptive field properties in thalamic neurons [8]. The electrophysiological signals (ongoing and sensory-related potentials and GFO) appear as valuable neurophysiological markers to test the functional state of neural networks. Such quantifiable bioelectrical markers might thus be a promising translational tool to develop innovative therapies designed to prevent the occurrence of psychotic disorders. In short, ketamine decreases the signal-to-noise ratio at least in the CT-TRN-TC system [59,61,320]. Ketamine also transiently disrupts the expression of long-term potentiation in the TC system [61], disorganizes action potential firing in rat prefrontal cortex [321], increases the firing in fast-spiking neurons and decreases it in regular spiking neurons [322] and disturbs sensory-related cortical and thalamic GFO. Dizocilpine (MK-801) is, like its derivative ketamine, a well-known non-competitive NMDAR antagonist with psychotomimetic properties leading to similar but more sustained effects than ketamine [60,61,62]. It addition, dizocilpine modulates the expression of numerous genes in cortical and subcortical structures [323].

In summary, neural GFO represent a translational bioelectric marker. It may be considered a potential prognostic and therapeutic hallmark for cerebral network disorders underlying psychotic symptoms. Such a quantifiable marker might be a promising translational tool for understanding the etiopathophysiological mechanisms of psychotic disorders and for developing innovative therapies. These include, for instance, noninvasive neurophysiological modalities such as TES, applied in at–risk mental state individuals to prevent the occurrence of first–episode psychosis and chronic forms of psychotic disorders.

## 6. Potential Mechanisms of TES

Little is known about the mechanisms underlying the clinical, acute and chronic effects of TES techniques, which are expected to re-establish the normal functional state in dysfunctional cortical-subcortical networks and/or to recruit compensatory networks. Whatever the technique and specific setting considered, it is difficult to perceive an integrated view of the mechanisms that are responsible for and contribute to the expected and the observed clinical effects. The possible mechanisms include genetic, molecular, cellular and systems pathways as well as long-lasting processes involving plasticity. The nerve cells are embedded in a conductive medium, the extracellular space, an important interface between the exogenous and endogenous electric currents and the excitable and non–excitable elements involved in information processing. In addition, before reaching the excitable cellular and subcellular elements, the TES-induced electric currents cross several types of barriers, including the cranium, the meninges, the vascular network and the glial tissue [324,325,326,327]. Also, the applied electric field has two components, one parallel and the other one perpendicular to the brain surface [328]. The strength of these two components determines the relative influence of TES on the excitability of the neural and non-neuronal elements. All these barriers, as well as the ongoing changes in the brain state, are a source of interferences with the electric field. Taken together, TES is expected to target a large number of neuronal and glial elements over large cortical and subcortical regions.

The clinical effects of TES and the underlying short- and long-term mechanisms principally rely on the electrode type and stimulation parameters (stimulus location, intensity, duration, polarity) [329,330]. The TES effects on brain structures are non-selective, state-dependent [331], and the strongest impact is not necessarily exerted in neural structures that are located below the electrodes [332]. The TES effects on the cellular and subcellular excitable elements depend on their geometry and on their spatial orientation in the electric field [332,333]. Both the TES effects and the underlying mechanisms lie on a continuum of effects ranging from the stimulation settings to the ongoing genetic, molecular, cellular and network dynamics. The mechanisms underlying the effects of TES are the subject of intensive research (for a review see: [33,119,334,335,336,337,338,339,340,341,342,343,344,345,346,347]). Our current knowledge remains fragmented with multiple and diverse proposed mechanisms: conduction block [348], synaptic potentiation or depression [349,350,351], network resonance [352], modulation of brain oscillations [127,337,353,354,355,356,357,358,359], of ongoing cellular firing and subthreshold membrane potential oscillations [360,361], of dendritic inhibition [362], of the astrocytic Ca^2+^/IP3 signaling [363] and of the synaptic efficacy in excitatory and inhibitory pathways [364]. It is reasonable to assume that multiple mechanisms are likely to operate in combination. The combination of these multiple mechanisms over time can be viewed as “meta-mechanisms” at the brain-network level.

In the following, I speculate on a few possible mechanisms that may, depending on the TES modality, be involved in the modulation of the layer VI CT pathway, which massively innervates both the dorsal thalamus and the TRN. As mentioned above, this glutamatergic pathway may be one of the prime targets for preventive TES in at-risk mental state individuals for psychotic disorders. In the present discussion, I take fundamental principles of neurophysiology into consideration. As the electrical field spreads at the speed of light, all neural and non-neural elements will be affected at the same time. The TES effects are expected to be attenuated with distance, obeying to the rule that the amount of current delivered by the electrode is proportional to the square of the distance between the brain elements and the stimulation electrode [365]. Figure 4 illustrates some of the anatomo-functional elements of the CT-TRN-TC system, which may somehow be impacted by TES electric fields.

TES entrains neuronal populations: Nerve cells operate on the basis of electrical charges, which makes them also responsive to weak electric currents [366,367,368,369,370]. Importantly, it was demonstrated that, in the rat, TES can directly entrain neurons in multiple neocortical areas and sub-neocortical structures, including the hippocampus [360]. Indeed, some of the cortical and hippocampal neurons were affected at similar phases of the TES oscillations, suggesting the contribution of non-synaptic mechanisms in the TES-induced direct entrainment of cortical and subcortical neurons. Of course, the directly activated neurons become a source for subsequent polysynaptic mechanisms, which represent a significant contribution in TES-induced entrainment over large cortical territories and the related subcortical structures. The percentage of TES phase-locked neurons depends on the state of brain networks and increases with TES intensity [360]. Furthermore, intracellular recordings revealed that both the firing and the underlying subthreshold and suprathreshold membrane potential oscillations are under the combined influence, through amplification, attenuation or interference, of TES fields and global network activities [360]. The mechanisms underlying the TES direct effects on ongoing neuronal activity are not well understood.

The axonal membrane, the more excitable element: The axonal membrane is generally more excitable than the somatic and dendritic membranes [371,372,373]. There is increasing evidence supporting the hypothesis that distal parts of the axon, remote from the axon initial segment, can autonomously integrate and generate action potentials, which could contribute to the emergence of field GFO involving synchronized GABAergic rhythmic activities [374,375,376,377,378]. So, it is predictable that the TES–induced electric field would create regional conditions in cortical tissue favorable for activating axons. Also, the number and the location of the activated axonal areas depend on both the neural tissue architecture and geometry, in relation to the direction of the electric field. The pattern of activated axons depends on the direction of the electric field and of the state of the cortical region being stimulated. The more numerous intersecting axons within an axonal bundle, the more numerous the axonal couplings [379]. An axon curvature would be as excitable as the initial segment [380]. Because the axonal membrane is more excitable than the somatodendritic membrane and is endowed with integrative properties [372,376,378], the TES-induced field may activate intracortical axons and axonal endings, where action potentials may be initiated. Dopamine and kainate can generate axon membrane depolarization leading to action potential initiation [381,382]. Also, oligodendrocytes, in addition to regulating myelination, would play a promoting role in synchronizing firing through axons [383]. Axo-axonal interactions can also involve glial cells [384]. Once triggered, ectopic axonal action potentials would run along the axons simultaneously both orthodromically up to the axon terminals and antidromically up to the parent somatodendritic domains. The orthodromically conducted action potentials would then activate local and distant postsynaptic neurons. A single orthodromically conducted action potential can even itself generate, for a while (a few tens of ms), a sequence of excitatory and inhibitory synaptic events in a subpopulation of interconnected glutamatergic and GABAergic neurons [385]. On the other hand, the antidromically conducted action potentials can activate directly the parent somatodendritic complex [376,378]. Moreover, TC neurons can spontaneously generate ectopic axonal action potentials, which subsequently modulate the parent soma’s excitability [376,386].

Cortical neurons are excited when the electric field is directed from the dendrites toward the axon [328]. The impedance mismatch between the dendritic arbor and the principal axon represents a likely mechanism for TES-induced cortical excitation [328]. Moreover, low-intensity electric fields concurrent to suprathreshold synaptic inputs can modulate the timing of action potential initiation [387]. Therefore, TES has the potential to influence the functional input-output balance in neurons.

Electrical couplings: Couplings between neurons in the central nervous system can occur through electrical synapses, that is, gap junctions [388,389]. They represent another potential target for the TES electric field. An important feature of these synapses is that they are bidirectional. Axo-axonal electric coupling via gap junctions is thought to contribute to the oscillating and integrative properties of neural networks [389,390,391,392,393,394]. This is a possible mechanism through which periodic TES can entrain oscillating neural networks. Sparse electrical couplings through axo-axonal gap junctions play a key role in the initiation and spreading of network gamma and higher (>100 Hz) frequency oscillations [392,393,395]. The coupling action potentials occur in the axon prior to invading the parent somatodendritic tree [389]. Such a mechanism represents a fast device for signal transmission directly between the outputs of neurons, thereby leading to temporally precise firing during fast network synchrony [389]. This supports the notion that the axon and its branches are not only reliable transmission cables for action potentials but also functional entities with integrative properties [376,396,397]. In the presented CT-TRN-TC system (Figure 4), it is shown that intracortical GABAergic parvalbumin-expressing interneurons communicate with each other through electrical and chemical synapses, which are functional modalities of tight couplings that contribute to the generation of synchronized oscillations in cortical structures [398]. Such couplings may also be influenced by TES.

Electrical couplings between central nervous system neurons can also occur through direct electrical (or ephaptic: touch phenomenon through ion exchanges between two adjacent excitable membranes) coupling [399,400]. Electrical couplings could also be mediated by the electric field generated jointly by many parallel axons. Such couplings might significantly be influenced not only by endogenous but also by exogenous electric fields independently of electrical and chemical synapses [401,402,403]. Therefore, ephaptic couplings may be one important target for TES. Ephaptic couplings between axons might be involved in the synchronization and the timing of action potentials as well. Endogenous or exogenous oscillating electric fields impose temporal windows, during which periodic ephaptically-induced membrane polarization can become the source of enhanced excitability in the corresponding neurons [401,404]. Thereby, ephaptic coupling leads to coordinated spiking activity among nearby neurons [401]. Ephaptic coupling influences the synchronization and timing of firing in neurons receiving suprathreshold synaptic inputs [360,387,405,406].

Combined digital and analog coding: It is usually taught that excitatory and inhibitory synaptic inputs modulate the integrative properties of the somatodendritic membrane areas, which lead to local voltage fluctuations (synaptic activity) that propagates up to the axon hillock, the non-myelinated segment of the principal axon, which will initiate a firing pattern (barcode) subsequently transmitted to the downstream synapses [407,408,409,410,411,412]. Once initiated, the action potentials simultaneously propagate orthodromically along the principal axon up to the presynaptic terminals, where they cause Ca^2+^ influx and transmitter release [413], and antidromically into the somatodendritic arbor, preventing the activation of the trigger zone at a proper time and/or triggering dendritic activities [414]. In vitro studies have demonstrated that, in the cerebral cortex and the hippocampus, somatodendritic voltage fluctuations can propagate over significant distances along the axon, change the amplitude and duration of the axonal action potential and, through a Ca^2+^-dependent mechanism, change the amplitude of the corresponding postsynaptic potentials [415,416]. In short, axons can transmit analog signals in addition to action potentials (Figure 4). Such a combined digital and analog coding represents an additional mechanism for information processing in neural networks. This dual coding must be a functional target for TES. It can be predicted that a TES-induced field can, for instance, modulate (via amplification, attenuation or interference) the amplitude of the voltage fluctuations running along the axon with subsequent impact on the action potential-evoked transmitter release at the corresponding synapses.

Top–down control: The first neural elements that are intensely impacted by any TES technique are, by all likelihood, first located in the more superficial layers of the cerebral cortex. The layer I or molecular layer, which is situated just underneath the pial surface, contains dense bundles of axons and dendrites and a paucity of sparsely distributed cell bodies [417,418]. Some of these axons give rise to descending axonal branches that innervate cortical neurons, thereby exerting a top-down control on the cortical and subcortical networks. For instance, it was demonstrated that the electric current generated by TMS can activate a network of GABAergic interneurons that innervate, in the upper cortical layers, the apical dendrites of layer V pyramidal neurons [362]. This GABAergic inhibition would be mediated by GABA(B) receptors, and their activation would decrease or suppress dendritic Ca^2+^ currents implicated in the synaptically-mediated dendritic excitation, which is involved in the integration of information. Even if such a scenario could also apply to the dendrites of layer VI CT neurons, it remains a challenge to predict, from the inspection of individual mechanisms, the actual activity pattern of the CT neurons. Assuming that TES inhibits their somatodendritic activity and firing, reduced firing of layer VI CT neurons, which exert a massive excitatory pressure on both TC and TRN neurons (Figure 2), would lead to a disfacilitation of the thalamic activity. So, the proposed TES-induced CT disfacilitation would reduce first the monosynaptic excitation of the glutamatergic TC and GABAergic TRN neurons and, secondly, the disynaptic inhibition of the TC neurons (Figure 4). Furthermore, because of the presence of dendrodendritic synapses between TRN neurons [168,171] and because of their pacemaker properties for GFO [232], TES-induced disfacilitation would also reduce the multisynaptic intra-TRN inhibitions [176], a possible brake for the generation of thalamic GFO. It is thus tempting to hypothesize that such TES-induced intracortical dendritic inhibition can reduce the power of GFO in cortical and subcortical structures.

Bottom–up effect from the thalamus: So far, TES has been presented as a noninvasive therapeutic means exerting top-down effects from the current source. Such effects can be categorized into at least two principal types of mutual interactions: local type with top-down controls, for instance the one discussed above, and a highly–distributed type, which involves interconnected cortical and subcortical networks. Indeed, it was well demonstrated that TES can directly, likely through non–synaptic mechanisms, entrain/modulate subcortical neurons [360]. The TES-induced electric fields would act as endogenous electric fields, which are known to guide network activity, to modulate the timing of action potentials [419], and to enhance stochastic resonance [420]. This is valid for both tDCS (static electric field) and tACS (alternating electric field) with effects on brain function [387]. The TES–induced field effects would modulate the amplitude of subthreshold and suprathreshold membrane potential oscillations of the target neurons. Because of a large number of variables mentioned above, it is difficult to provide a precise picture of the direct effects of TES field on both GABAergic TRN and glutamatergic TC neurons. Whatever the differential effects, both types of neurons work together and mutually influence each other. TES would affect their threshold mechanisms equivalent to an integrate–and–fire model, which depends on a certain number of factors, including the ion channel kinetics, the weight of excitatory and inhibitory synaptic inputs and the shape of the membrane potential distribution near threshold [421,422]. However, as mentioned above, we should keep in mind that both TRN and TC neurons generate action potentials during sustained membrane potential depolarization and hyperpolarization.

The GABAergic TRN neurons can be considered a privileged cell-to-network target for TES indirectly through the CT pathway, as mentioned above, and directly. In the following, I will speculate on possible mechanisms underlying eventual direct effects at the thalamic level, more specifically in the GABAergic TRN cells. It is first important to know that, at a sufficiently hyperpolarized membrane potential, TRN cells are more excitable and electro-responsive than TC neurons. Indeed, high-frequency bursts of action potentials generated by excitatory inputs require a higher degree of convergence of excitatory inputs than TRN neurons [423]. Furthermore, TRN cells are endowed with low-threshold T-type calcium currents of longer duration than TC neurons [424,425]. These anatomofunctional properties suggest that TRN cells may be more electro-responsive to TES than TC neurons. Gap junctions-mediated electrical synapses is another important characteristic of GABAergic TRN neurons [172,426]. Such electrical synapses are implicated in diverse forms of cell-to-network signaling. Using a novel dye-coupling technique, Connors and colleagues [427] further demonstrated that, in the rodent, electrically coupled TRN cells form clusters with distinctive patterns and axonal projections. Unpredictably, TES would facilitate the synchronized generation and spread of electrically- and chemically-induced synaptic activities within TRN clusters. The presumed TES-induced TRN activity patterns would strongly influence network oscillations, which would generate inhibitory activities (principally lateral inhibition [179]) in the related populations of TC neurons. To sum up, alternating TES (or tACS) is expected to influence directly the thalamus, which is a well-known oscillator [221]. Here, in the present conceptual context, the thalamus is a reference. This means that the proposed mechanisms underlying direct subcortical effects could apply to other sub-neocortical structures, such as the hippocampus [360], along with their respective anatomofunctional properties.

Another direct influence relies on the fact that, as above mentioned, the axonal membrane is more excitable than the somatodendritic membrane. So, assuming that the ongoing state of the cortex allows TES-mediated triggering of action potentials on axonal terminations of TC neurons, the corresponding ectopically-generated axonal action potentials would backpropagate up to the parent somatodendritic complex of these TC neurons. Such antidromically conducted axonal action potentials would influence the somatodendritic excitability of the corresponding TC neurons [376]. If true, such an effect may, under suitable circumstances regarding the network state, short-circuit the CT-mediated thalamic multisynaptic effects. In theory, such an effect would be more efficient when the somatodendritic field is hyperpolarized (see Figure 35 in [376]). In short, TES would not only generate action potentials on ectopic axonal membrane but also modulate the timing of action potential initiation in the axonal and somatodendritic membrane [387], thereby influencing the dynamics and plasticity of neural networks.

In summary, regarding its multiple and diverse mechanisms, TES would exert local and highly distributed influences on the ongoing thalamic activities through at least three principal ways. They would, over time, occur individually or in combination leading to polysynaptic effects (Figure 4): (1) Direct, intracortical synaptic and non-synaptic (especially electrical) mechanisms, thereby modulating the excitability of the CT axon (from the hillock to the ascending ramifying axon collaterals) and the integrative properties of the CT neurons’ somatodendritic arbor; (2) direct, intracortical modulation of the excitability of TC axon terminals, which could initiate antidromic firing along the principal axon of TC neurons with subsequent effects on the TC neurons’ somatodendritic membrane state-dependent excitability and a monosynaptic excitation of TRN cells; (3) direct forced TES field effect on thalamic neurons, especially on the GABAergic TRN cells because they are endowed with more “explosive” electrophysiological properties than TC neurons, leading for instance to phase–biased cellular firing. Whatever the TES-induced top-down and bottom-up (from presumed direct effect on thalamic neurons) mechanisms, the effect on the excitability and integrative properties of all the elements (including non-neuronal) that make up the CT-TRN-TC system would depend on its ongoing functional state. At any rate, TES-elicited modulation of the CT pathway should influence the thalamic neurons’ spatiotemporal properties, which are related to the center-surround receptive field [428]. The predictions and hypotheses presented in the present essay need to be validated through appropriate experiments.

## 7. Conclusions and Perspectives

In the present essay, I began with a neurophysiological perspective on early therapeutic intervention (TES) in at-risk mental state individuals against the occurrence of FE psychosis, chronic psychotic disorders, and schizophrenia. Because of their noninvasiveness, low-cost and safety, the use of TES therapeutic modalities, which are almost free of side-effects, is increasing over years with encouraging and promising clinical outcomes. Furthermore, there is accumulating evidence that static (DC field) or alternating (AC field) TES exerts an effect on brain function. On the other hand, the underlying mechanisms still remain elusive. There is accumulating evidence that exogenous electric currents can modulate not only brain electrical activity but also behavioral and cognitive performance. All the proposed mechanisms belong to a continuum that can be considered “meta-mechanisms” at the brain-network level. The use of TES may be seen as a “natural” treatment as it can influence, like endogenous electric fields, the excitability and the integrative properties of the brain nerve cells and subcellular elements. TES can, through the extensive CT and cortico-cortical systems, nonselectively affect, directly and through multisynaptic pathways, global brain activity. This is not surprising since electrical modulation of a single neuron can modify not only the global brain state [429] but also motor behavior [430], and that a single action potential can itself generate sequences of excitatory and inhibitory synaptic events in subnetworks [385].

On the basis of our current knowledge, it is tempting to put forward that noninvasive therapeutic interventions using TES might turn out to be very promising in the future as there is emerging evidence that TES might supplant surgical DBS therapy against neurobiological disorders, including Parkinson’s disease. This might also be the case for epilepsy, dystonia, obsessive compulsive disorders, pain, multiple sclerosis, addiction, depression, Tourette syndrome, and in brain–injured patients in vegetative and minimally conscious states. That TES (with settings adjusted on the basis of cortical-subcortical oscillations) can be applied to treat any neurobiological disorder rests on the notion that TES would set into action highly-distributed networks, which would help the brain, in case of dysfunctional networks associated with disturbed oscillations, to retrieve its fundamental capability to self-organize, self-calibrate and self-correct [431,432].

In the present essay, there is a bias toward rodent models merely because the CT-TRN-TC system of rodents and humans share common anatomo-functional properties. The CT-TRN-TC system was put on the stage since the widespread CT pathway, the thalamus and the glutamatergic synaptic transmission together form an etiopathophysiological backbone for schizophrenia and, therefore, may represent a prime therapeutic target for preventive TES of dysfunctional brain networks in at-risk mental state individuals against the occurrence of first-episode psychosis and schizophrenia. Importantly, the one common denominator is cortical GFO, which are amplified not only during hallucinations but also in at-risk individuals for psychosis and during first-episode psychosis. Furthermore, abnormal network gamma hypersynchrony is likewise recorded in the CT systems of healthy humans and rodents after a single systemic administration, at a psychotomimetic dose, of the NMDAR antagonist ketamine. These translational ketamine models of prodromal to first-episode psychosis are thus promising means not only to work out a preventive treatment against the occurrence of chronic schizophrenia but also to investigate the TES mechanisms.

An important question remains open as to whether TES is capable of replacing the generalized cortical-to-subcortical ongoing gamma hypersynchrony for normal gamma synchrony. Importantly, alpha frequency (10 Hz) tACS can reduce the power of cortical GFO [433]. This is interesting since ongoing alpha oscillations are often associated to cortical idling whereas GFO are associated to an attentional state [128]. Because of its activity-dependent dynamic properties, the CT pathway is expected to play a crucial role in the modulation of the ongoing activity in the CT-TRN-TC system. Indeed, a comprehensive in vitro study performed in rodent CT-TRN-TC slices demonstrated that a low-frequency optogenetic stimulation (≤10 Hz) exerts a suppressive effect on TC neurons’ activity [205]. In other words, we predict that TES of the extensive CT pathway can re-normalize/improve its key attentional role to generate in the dysfunctional CT-TRN-TC circuits, more specifically in at-risk mental state individuals, “normal” prediction models that guide the flow and sequences of sensorimotor and cognitive processes. Such mechanisms would operate in combination with the TRN, strategically located at the interface between the dorsal thalamus and the neocortex, through the modulation of the excitatory and suppressive components of the receptive fields in the appropriate and related cortical and thalamic territories. Thereby, principally under the influence of the TC pathway, the TRN may fine-tune the responsiveness of sensory, motor and cognitive TC neurons, depending on the ongoing functional brain state and on the relative timing of the multiple and diverse thalamic inputs.

Now, if TES were able to suppress the psychosis-relevant CT-TRN-TC gamma hypersynchrony, thought to be the electrophysiological correlate of a hyper-attentional state, would it stop the occurrence of the schizophrenia-relevant clinical disorganization and the emotional, sensorimotor and cognitive abnormalities? This is a fundamental issue that certainly needs further investigation.

Also, should TES be systematically applied, along with ethical guidelines, in a standard fashion to all at-risk mental state patients for psychotic disorders? Probably not. This is an important issue because its efficiency depends on multiple variables, more specifically on the brain state and longitudinal outcomes. Also, in an attempt to effectively apply TES at the right time, it might be necessary to use a closed-loop feedback system able to trigger the stimulation on the basis of the pattern of the ongoing brain activity [434].

On the other hand, TES may be supplanted by, or combined with, other non-pharmacological therapies, for instance, with cognitive remediation and psychotherapies. These latter therapies are promising when it comes to helping individuals with impaired cognitive performance [146,435,436,437]. Mindfulness-based therapy may also be an interesting alternative or complement to TES [438]. All these alternatives mean that a good quality of life prevails for at–risk mental state and first-episode psychosis patients [44] with or without a rational use of TES in the frame of a personalized medicine.

## Figures and Tables

**Figure 1 brainsci-07-00034-f001:**
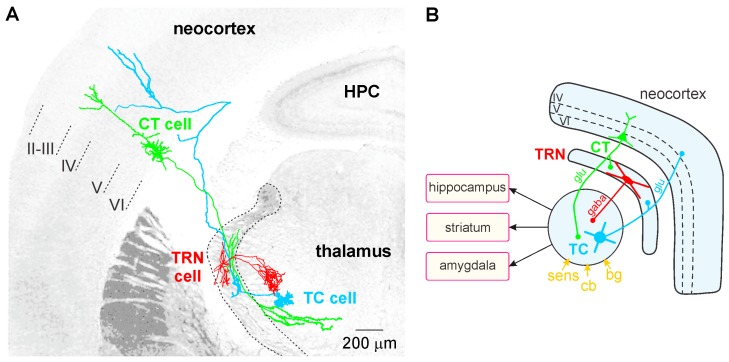
The principal anatomical features of the rodent cortico-reticulo-thalamocortical (CT-TRN-TC) system. This is the principal CT-TRN-TC system that is common to first- and higher-order thalamic nuclei. (**A**) Mounting of reconstructed juxtacellularly-labelled neurons of the rat primary somatosensory system. Both the CT (in green) and the TC (in blue) neurons are glutamatergic (glu) and their principal axon crosses the TRN where it gives rise to axon collaterals. The TRN neuron is GABAergic (gaba) and innervates only the TC neurons of the dorsal thalamus principally through lateral inhibition. (**B**) In this schematic drawing of this 3-neuron circuit, the principal afferents (bg, basal ganglia; cb, cerebellar; sens, sensory) and efferents of the dorsal thalamus are indicated, the TRN being part of the ventral thalamus.

**Figure 2 brainsci-07-00034-f002:**
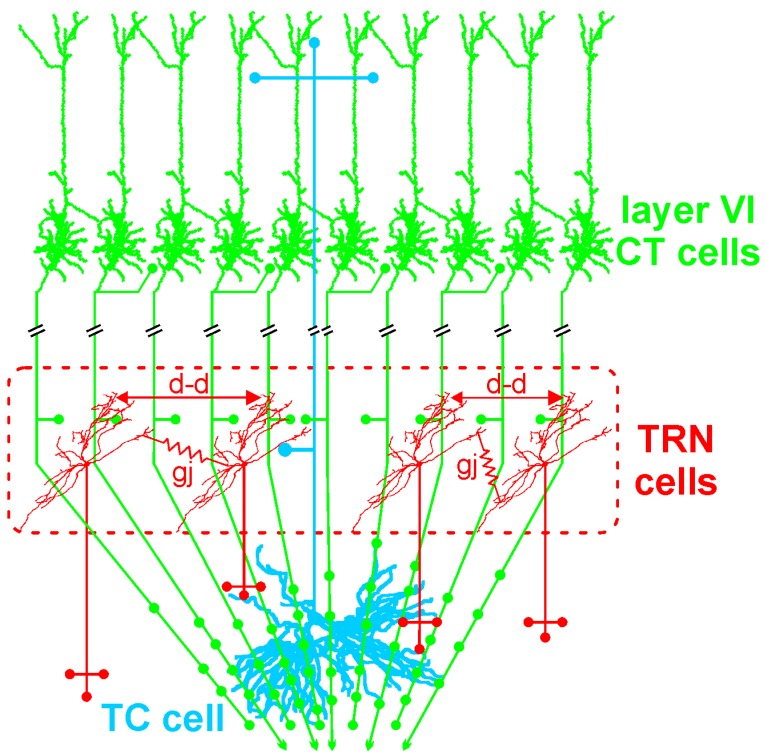
The layer VI corticothalamic (CT) neurons outnumber the thalamocortical (TC) neurons by a factor 10. As a consequence, the glutamatergic CT neurons exert a widespread and powerful excitatory influence on the first- and higher-order thalamic nuclei. Layer VI CT axons innervate other layer VI CT neurons via recurrent axon collaterals. In contrast, the glutamatergic TC neurons do not communicate among each other. The GABAergic TRN cells use dendro-dendritic chemical (d-d) and electrical (gj, gap-junction) synapses to communicate among each other.

**Figure 3 brainsci-07-00034-f003:**
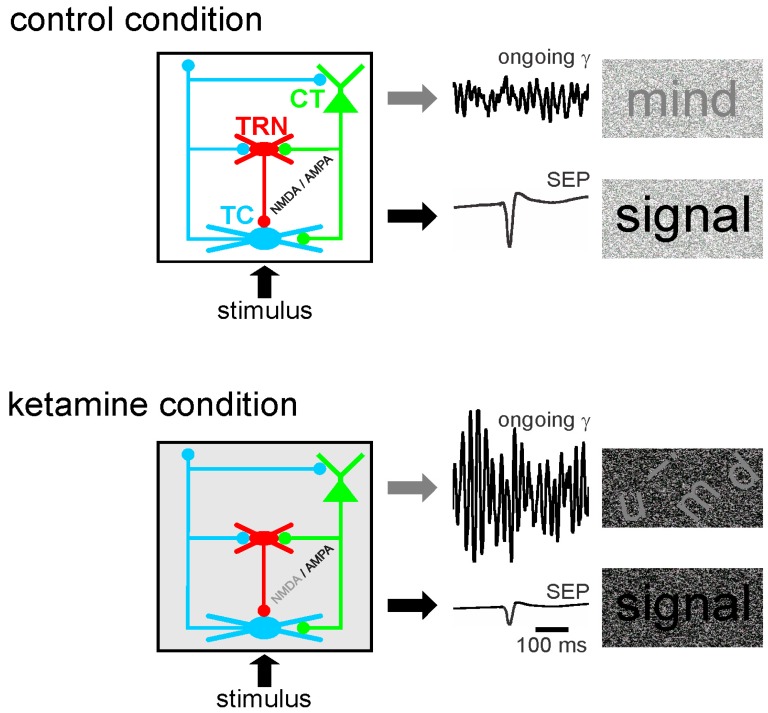
The NMDAR antagonist ketamine decreases the ability of the cortico-reticulo-thalamocortical (CT-TRN-TC) system to integrate incoming information. A single systemic administration of ketamine disturbs the functional state of the three-neuron circuit (layer VI CT-TRN-TC). Ketamine increases the amplitude of spontaneously occurring gamma frequency oscillations and decreases the amplitude of the sensory-evoked potential in both the thalamus and the neocortex. Layer VI CT neurons innervate the thalamic relay (TC) and reticular (TRN) neurons through the activation of glutamate ionotropic (NMDA and AMPA) and metabotropic receptors. Ketamine is expected to decrease the NMDA/AMPA ratio at least at CT synapses. Thereby, ketamine disturbs the mental state and decreases the gamma signal–to–noise ratio in the CT-TRN-TC system. The sensory-evoked potential (SEP) can be considered as an index of the sensory-related signal. Adapted from [61] and from [59].

**Figure 4 brainsci-07-00034-f004:**
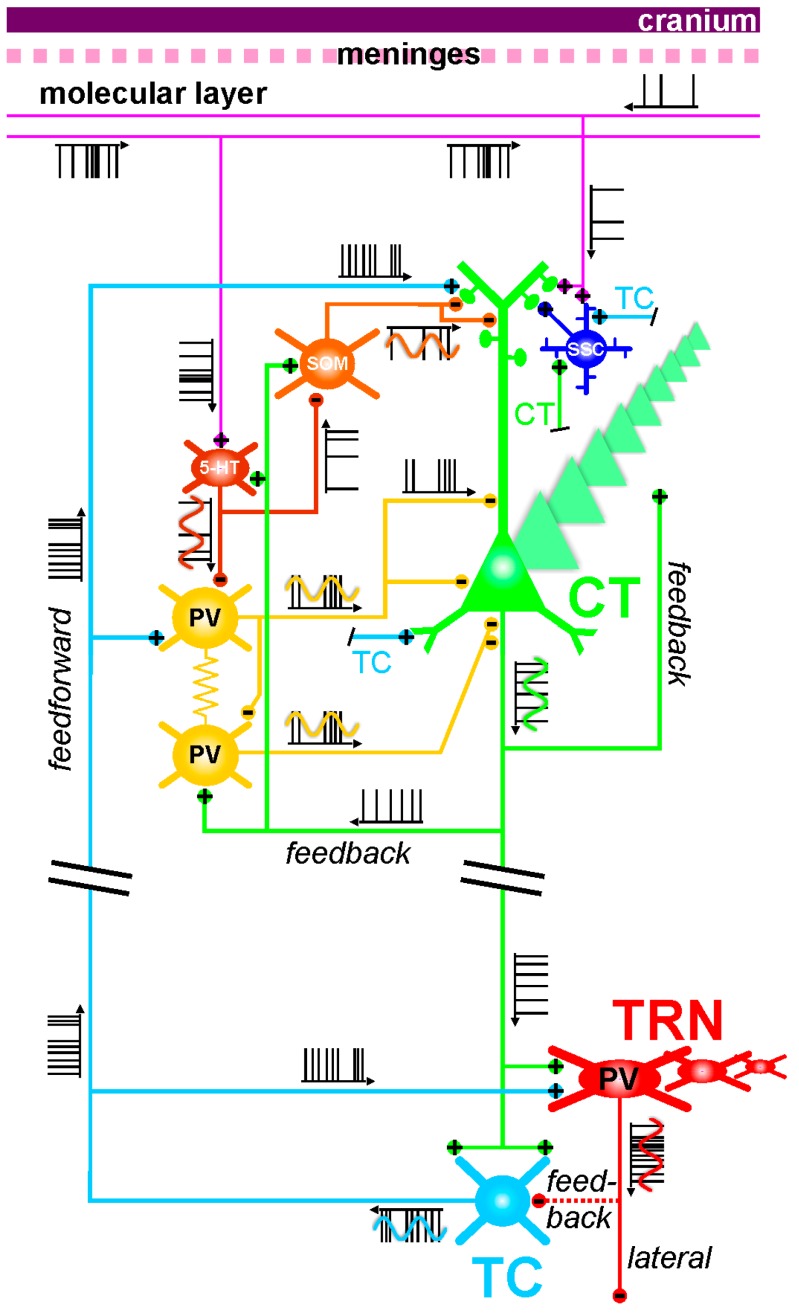
Potential mechanistic targets in the cortico-reticulo-thalamocortical (CT-TRN-TC) system for transcranial electrical stimulation. This model includes three parts, which are assumed to work together: (i) The innervation of the intracortical circuitry by both the descending axonal branches (top-down process) of the axons running within the molecular layer and the ascending TC inputs (bottom-up process); (ii) functional interactions between glutamatergic and GABAergic neurons of the intracortical circuitry, which includes feedback and feedforward excitations (from CT and TC axon collaterals, respectively); and (iii) the layer VI CT pathway, one of the outputs of the intracortical circuitry, which innervates simultaneously the thalamic GABAergic reticular (TRN) and glutamatergic relay (TC) neurons. In this model, the TRN cells generate more lateral than feedback inhibition in the dorsal thalamus, which contains only TC neurons. The layer VI CT axonal projections are about ten–fold higher in number than the TC projections, thereby generating a great excitatory pressure on TRN and TC neurons. Furthermore, the apical dendrites of layer VI pyramidal neurons terminate in layers III–IV. Each neuron exhibits its own firing pattern that is state-, voltage-, synaptic- and time-dependent. The action potentials (APs) are drawn like a code bar. Under physiological condition, it is assumed that the APs are initiated at the axon hillock, the initial segment of the axon. The axon can also transmit, in addition to APs, analog signals (generated in the somatodendritic domain and represented by sinusoidal waves) along the axon (at least several hundreds of micrometers away from the soma) and can modulate AP-evoked transmitter release at the corresponding synapses. In this model, it is assumed that axodendritic (chemical synapses) and dendrodentritic electrical (via gap junctions) coupling exist between the two types (basket and chandelier) of GABAergic parvalbumin (PV) expressing cells. 5-HT, 5-HT3A receptor; CT, corticothalamic; SOM, somatostatin; ssc, spiny stellate cells; TC, thalamocortical; TRN, thalamic reticular nucleus.

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
