# Peer review of "A Neurophysiological Perspective on a Preventive Treatment against Schizophrenia Using Transcranial Electric Stimulation of the Corticothalamic Pathway"

_brainsci, 2017, doi:10.3390/brainsci7040034_

Round 1

Reviewer 1 Report

A well developed paper on the difficult topic of mechanisms involed in TES. The paper presents a theoretical proposal as to how we may understand the mechanisms of TES. The proposal is well constructed and justified.

Author Response

Dear Reviewer:

I thank you very much indeed for your positive feedback.

The revision was made under track mode. The manuscript has been revised, in an attempt to make it more comprehensive, and polished then checked by a native English speaking colleague.

With my best regards

Didier Pinault

Reviewer 2 Report

Both, the proposed hypothesis, i.e. non-invasive brain stimulation as a preventive treatment strategy in the context of schizophrenia as well as the review of concepts of schizophrenia and principles of non-invasive brain stimulation are for themselves convincing though detached from one another.

The manuscript is hard to follow, little structured and hardly discriminates between redundant information, suppositional knowledge and new insights and interpretations.

A few examples on this criticism:

-          Paragraph on Tourette is arbitrary in this context

-          Paragraph on DBS seems redundant in special issue context

-          Paragraph on animal models of schizophrenia is not leading anywhere also give the fact that the most widely used, validated and reviewed models of schizophrenia are not cited or reflected. In contrast, the proposed/defended ketamine model does not constitute a developmental model essential to study preventive interventions in schizophrenia

-          Paragraph on thalamus is by far too complex, a reduction to its involvement in schizophrenia pathology might be sufficient

-          The justification for application of brain stimulation approaches in schizophrenia pathology based on its successful application in neurological disorders such as tremor is not seminal

-          In the context of a neurophysiological perspective on preventive non-invasive treatment of schizophrenia extensive elaborations on a potential anatomical target for advanced schizophrenia is counterproductive given the neurodevelopmental and neuroprogressive nature of schizophrenia

-          The mention of ECT in the present context is redundant

-          In contrast a paragraph on how the author speculates non-invasive stimulation approaches that only have a restricted current depth (in)directly affects the thalamus is missing

Author Response

Dear Reviewer:

I thank you very much indeed for your constructive comments and suggestions. The revision was made under track mode.

All the points raised have been addressed (see below). In addition, the manuscript has been checked by a native English speaking colleague.

-Reviewer: Both, the proposed hypothesis, i.e. non-invasive brain stimulation as a preventive treatment strategy in the context of schizophrenia as well as the review of concepts of schizophrenia and principles of non-invasive brain stimulation are for themselves convincing though detached from one another.

 The manuscript is hard to follow, little structured and hardly discriminates between redundant information, suppositional knowledge and new insights and interpretations.

A few examples on this criticism:

-Paragraph on Tourette is arbitrary in this context

-Author: This para has been deleted.

-Reviewer: Paragraph on DBS seems redundant in special issue context

-Author: It has been deleted.

-Reviewer: Paragraph on animal models of schizophrenia is not leading anywhere also give the fact that the most widely used, validated and reviewed models of schizophrenia are not cited or reflected.

-Author: The goal of the present survey was not to develop on animal models but, in the present context, it was important to stress that all existing models are questionable and I cited at least 4 critical papers (ref: 68-71) on this topics. I also added a few well-known critical references on this topics (ref: 75-76). Here, the point is to mention that we need animal models for the development of “therapeutic concepts and to understand the neural mechanisms of electrical neuromodulation used in diverse interventions."

-Reviewer: In contrast, the proposed/defended ketamine model does not constitute a developmental model essential to study preventive interventions in schizophrenia.

-Author: The referee is right that the ketamine model is not a neurodevelopmental model. However, as defended here, on the basis of the current literature it is well-recognized to model more acute than chronic schizophrenia. In addition, there is accumulating evidence that the ketamine model models more at-risk mental state and first-episode psychosis than chronic schizophrenia. This is the reason why I think the ketamine model promising to study preventive intervention and I provided strong arguments for this perspective.

-Reviewer: Paragraph on thalamus is by far too complex, a reduction to its involvement in schizophrenia pathology might be sufficient

-Author: I do not fully agree with the reviewer:

1)    Layer V and layer VI CT have distinct anatomofunctional features. In the current literature there is a large number of papers referring especially to layer V CT neurons. Layer VI CT neurons are considered as more neuromodulator than layer V CT neurons (see sherman, Guillery). Of importance, only the layer VI CT neurons innervate both the thalamic relay (TC) and reticular (TRN) neurons, a reason that lead me to consider the layer VI CT pathway as a potential therapeutic target.

2)    The TRN is an essential structure in the functioning of the CT-TC system, and it plays a crucial role in thalamic oscillations, which are modulated by CT neurons.

3)    However, I decided to delete the para on the other four GABAergic inputs, which are not used in the following but which would certainly play certain roles when stimulating the cerebral cortex.

-Reviewer: The justification for application of brain stimulation approaches in schizophrenia pathology based on its successful application in neurological disorders such as tremor is not seminal.

-Author: I agree with the reviewer, so I amended the para accordingly. Indeed it is known that direct electrical modulation of the brain can modulate behavior since the 19th century.

-Reviewer: In the context of a neurophysiological perspective on preventive non-invasive treatment of schizophrenia extensive elaborations on a potential anatomical target for advanced schizophrenia is counterproductive given the neurodevelopmental and neuroprogressive nature of schizophrenia.

-Author: The reviewer raises an interesting concern, which implies a couple of disputable points.

1) Here it is not question to apply TES of the CT pathway in patients with advanced schizophrenia but why not on the basis of longitudinal clinical investigation.

2) The proposed perspective is on possible applications of TES of the CT pathway in at-risk mental state individuals against FE psychosis and chronic Sz.

3) An important question that is not developed in the present essay is whether TES of the CT pathway (frontoparietal or parietotemporal cortex) can prevent the development of advanced chronic schizophrenia, which is under influence of chronic treatments with serious side effects that may also impact positively and/or negatively the neurodevelopment of advanced chronic schizophrenia.

I amended the text (track mode), including the conclusion/perspective section in an attempt to address this important issue raised by the reviewer.

-Reviewer: The mention of ECT in the present context is redundant

-Author: I deleted the para on ECT

-Reviewer: In contrast a paragraph on how the author speculates non-invasive stimulation approaches that only have a restricted current depth (in)directly affects the thalamus is missing.

-Author: Thanks to the reviewer to give me the challenge to further develop the proposed neurophysiological perspective. So I added a subsection entitled: “Bottom-up effect from the thalamus”.

With my best regards,

Didier Pinault

Round 2

Reviewer 2 Report

Though the ms has somehow improved by the author addressing most of the concerns, I am still not convinced by the animal model chosen and don’t see the necessity to stress the pharmacological acute ketamine model.

The author might be aware of the growing literature on preventive interventions in maternal immune stimulation models of schizophrenia that mimic both behavioral and neurobiological abnormalities relevant to schizophrenia as well as the developmental nature of schizophrenia reflected in a significant maturation delay. I wonder why the author persists on the ketamine model on the one hand and to disregard this most widely studied animal model of schizophrenia on the other hand and in a context where exactly this, the developmental maternal immune activation model is highly significant and relevant.

Schizophrenia by nature is not acute, neither is its first episode. Preventive interventions need to consider the chronic nature as well as the development from at-risk mental state and first-episode psychosis to chronic disease expression. It goes without saying that animal models of psychiatric disorders are questionable. This well accepted notion makes it even more important to select a model optimal for the selected investigatory approach. Preventive interventions require developmental models as even a preventive intervention will meet a pathological brain and intervention act pathology-dependent.

As both, the proposed hypothesis, i.e. non-invasive brain stimulation as a preventive treatment strategy in the context of schizophrenia as well as the review of concepts of schizophrenia and principles of non-invasive brain stimulation are for themselves convincing, I wonder whether stressing the ketamine model with all its significant limitations is essential for the ms and whether a small outlook on the necessity of animal experimental studies to prove the concept wouldn’t be enough.

In summary, I would like to see the ketamine model taken out or stressed less as it rather reduces than it adds to the significance and expressive power of the ms.

Author Response

I thank very much the reviewer for the positive assessment of my ms and especially for her/his pertinent and constructive comments and suggestions, which help me to improve the ms.

Why “stressing” the acute pharmacological acute ketamine model?

From the current literature the acute ketamine translational, human and non-human animal models are used to test the NMDAR hypofunction hypothesis of schizophrenia (references in the ms). It is well established that these models model more acute than chronic forms of schizophrenia, more specifically ARMS for psychosis and first-episode psychosis. In the present ms, I posit that the acute ketamine model is an appropriate means, based on the NMDAR hypofunction hypothesis of schizophrenia, to validate a proof-of-concept for therapeutic neurophysiological interventions in ARMS against FE psychosis and chronic schizophrenia.

The present ms does not, anyway, negate the other models of/for schizophrenia. Indeed, I well recognize the prenatal immune stimulation models which, nowadays, may be the best neurodevelopmental models of schizophrenia, and which can be used to prevent the development of schizophrenia-relevant behavioral and neurobiological abnormalities (Meyer, Prog NPP Biol Psychiatry, 2013; Giovanoli et al., Transl Psychiatry, 2016). However, it is worth specifying that, in such neurodevelopmental models, the preventive treatment (antibiotic) targets one of the causal factors, that is, a stress-induced inflammatory process that would be responsible for the schizophrenia-relevant observed abnormalities. This type of treatment is presymptomatic. It would absolutely be terrific if we were able to treat all at-risk individuals for psychosis, for instance using such a preventive presymptomatic treatment.

To the best of my knowledge, there is a real need to treat high-risk patients for whom the causes of their mental state remain unknown. This is the heart of my review in which, on the basis on the NMDAR hypofunction hypothesis of schizophrenia, I propose a neurophysiological perspective on a preventive treatment against this chronic psychotic disorder. This type of treatment would be more symptomatic than pre- or asymptomatic, and it would require reliable bioelectrical markers of brain network states (e.g, EEG oscillations). The goal of the present essay is also to understand the potential mechanisms, focused on the CT system, underlying the TES effects.

Therefore, I amended the introduction and the subsections starting by “Animal models” (last 2 para of section 2) along these lines